# SESS Model for Adolescent Sexual Health Promotion: A Quasi-Experimental Two-School Evaluation in Thailand

**DOI:** 10.3390/ijerph22101536

**Published:** 2025-10-08

**Authors:** Jun Norkaew, Pissamai Homchampa, Souksathaphone Chanthamath, Ranee Wongkongdech

**Affiliations:** 1Faculty of Medicine, Mahasarakham University., Maha Sarakham District, Maha Sarakham 44000, Thailand; jun_nor@vu.ac.th; 2Faculty of Public Health, Vongchavalitkul University, Nakhon Ratchasima District, Nakhon Ratchasima 30000, Thailand; 3Faculty of Nursing, Burapha University, Chonburi District, Chonburi 20131, Thailand; pissamaih@gmail.com; 4Khammouane Provincial Hospital, Thakhek 9RQ4+VQ2, Laos; souksathaphonec@gmail.com; 5International and National Collaborative Network and Innovation for Community Health Development Research Unit (INCNI-CHD), Mahasarakham University, Maha Sarakham 44000, Thailand

**Keywords:** sexual health, adolescents, school-based intervention, empowerment, social support, quasi-experimental study

## Abstract

**Background:** Unintended adolescent pregnancy and sexually transmitted infections (STIs) remain pressing public health concerns in Northeastern Thailand. Although school-based sexuality education is widespread, risk behaviors persist, underscoring the need for innovative approaches. This study evaluated the SESS (System–Empowerment–Support–Social Network) model, a multi-component framework designed to strengthen adolescent sexual health. **Methods:** A quasi-experimental, two-school study was conducted among 240 students aged 15–19 years in Nakhon Ratchasima Province. One school (*n* = 120) implemented a 16-week SESS program, while a comparison school (*n* = 120) continued with the standard curriculum. The SESS model combined system coordination, empowerment workshops, peer and institutional support, and digital platforms (Facebook, LINE). Data were collected with validated questionnaires and analyzed using ANCOVA, adjusting for baseline values. Exploratory analyses reported mean differences with 95% confidence intervals (CIs). **Results:** Groups were comparable at baseline. Post-intervention, the intervention school showed higher perception scores (mean difference = +13.0; 95% CI: 10.5–17.0) and preventive practice scores (mean difference = +14.0; 95% CI: 10.1–17.9). Attitudes showed minimal change. No pregnancies or self-reported STI cases were documented among intervention participants during the follow-up period. **Conclusions:** In this two-school quasi-experimental evaluation, the SESS model was associated with improvements in perceptions and practices, though attitudinal changes were limited. Findings suggest the feasibility of integrating empowerment, social support, and digital engagement into school-based programs while highlighting the need for multi-school trials to establish effectiveness.

## 1. Introduction

Adolescent sexual and reproductive health is a global public health concern. Each year, an estimated 21 million girls aged 15–19 become pregnant, with around 12 million giving birth—often unintentionally or under unsafe conditions [1]. According to the World Health Organization (WHO), youths aged 15–24 account for a substantial portion of over 1 million new sexually transmitted infections (STIs) acquired daily [2]. These figures underscore the urgent need for effective interventions, particularly in regions experiencing high rates of teenage pregnancy and STIs.

In Thailand, the Northeastern region reports some of the country’s highest adolescent birth and STI rates. In 2018, 9999 STIs cases were reported nationwide, with adolescents aged 15–24 accounting for more than 60% [3]. The rising incidence of gonorrhea and other STIs among youth further highlights the scale of the issue [4].

Although school-based sexuality education exists in Thailand, its impact has been limited. Studies show that curricula often focus narrowly on the negative consequences of sex, such as pregnancy and disease, while omitting important topics like consent, emotional health, and gender identity [5]. Many educators lack training and rely on lecture-based methods rather than participatory or student-centered approaches [6]. As a result, adolescents often turn to peers or unreliable online sources for information, thereby increasing their vulnerability to misinformation and engaging in risky behavior [7].

In response, the Thai government enacted the Prevention and Solution of the Adolescent Pregnancy Problem Act (2016), mandating comprehensive sexuality education and adolescent-friendly health services [8]. However, a significant gap remains between policy and practice, especially at the local level. Challenges include fragmented coordination among schools and communities, as well as limited awareness of youth rights among service providers [9].

Evidence suggests that multidimensional, theory-based programs combining knowledge, skill-building, empowerment, and social support are more effective than traditional approaches [10]. Digital engagement via social media and online platforms is also promising for reaching youth in culturally relevant ways [11].

This study aimed to develop and evaluate the SESS model—a school-based intervention grounded in System Theory, Empowerment Theory, Social Support Theory, and digital engagement. The SESS model (System–Empowerment–Support–Social Network) integrates system-level coordination, peer-led learning, youth empowerment, online support, and rights-based sexuality education tailored to the Northeastern Thai context.

The theoretical foundation emphasizes both individual agency and environmental support. Empowerment Theory, as articulated by Zimmerman, emphasizes the importance of equipping youth to make informed decisions about their well-being [12]. Social Support Theory emphasizes the significance of peers, families, and institutions in promoting protective behaviors during adolescence [13]. Digital strategies have also been shown to enhance sexual health knowledge and engagement among young people [14,15,16].

## 2. Materials and Methods

### 2.1. Study Design and Setting

This study employed a research and development (R&D) framework comprising three phases, followed by a quasi-experimental, two-school pre–post study with non-equivalent comparison groups. The design was originally conceptualized as a clustered randomized controlled trial; however, with only one school per arm, the study is more accurately classified as a quasi-experimental design. Data were collected between August 2021 and May 2022 in secondary schools in Nakhon Ratchasima Province, Thailand—a province purposively selected due to its high adolescent birth rates and elevated prevalence of sexually transmitted infections (STIs) among individuals aged 15–19 years.

### 2.2. Participants and Sampling

A total of 240 adolescents aged 15–19 years were recruited using a multi-stage sampling method. Eleven secondary schools in Nakhon Ratchasima Province were initially screened for eligibility. Two schools were purposively selected and matched on size, geographic area, and student demographics. One school was designated as the intervention site, while the other served as the comparison site.

Inclusion criteria were: (1) adolescents aged 15–19 years currently enrolled in secondary school; (2) ability to provide informed assent; (3) written informed consent from a parent or legal guardian; and (4) physical and mental capacity to participate in classroom and online activities throughout the 16-week intervention.

Exclusion criteria were: (1) diagnosed cognitive impairments or psychiatric disorders reported by teachers or guardians that could interfere with participation; (2) school transfer during the study period; and (3) refusal to participate.

The selection of the intervention school also considered institutional readiness, including the presence of active student health clubs, a supportive school administration, and the availability of school-based health services. The comparison school followed the standard Ministry of Education curriculum for sexuality education. To reduce contamination, the two schools were located in different catchment areas with no overlap of teachers or student networks.

### 2.3. Development of the Intervention: The SESS Model

Phase 1: Situational Analysis

Phase 1 was informed by a prior case–control study conducted by our research team among adolescent girls aged 15–19 years in Northeastern Thailand. Quantitative data were collected using structured surveys, and qualitative data through focus group discussions with students, school staff, and public health officers. The study, which has not yet been published, identified low sexual health literacy, limited parent–child communication, peer influence, and online misinformation as key risk factors, while family support and rights awareness were protective.

Phase 2: Model Development and Testing

Using action research over 16 weeks, two key components were implemented:

(1) Social Networks and Support Systems

School-based networks involving educators, health professionals, and student leaders co-developed sexual health strategies.Online support platforms were co-managed to disseminate reliable sexual health content and foster peer learning.

Empowerment Program Implementation

Stakeholder workshops co-designed a curriculum aligned with adolescent needs.Activities involved role-play, skill-building, problem-solving, and reflection, focusing on sexuality education, health promotion, pregnancy prevention, and rights.

### 2.4. Data Collection and Instruments

#### 2.4.1. Data Collection

Consent and assent procedures were followed by the ethical guidelines for research involving minors. Information sheets and consent forms were distributed in advance through schools, and signed parental/guardian consent was required for all participants under the age of 18, in addition to adolescent assent.

#### 2.4.2. Instruments

Quantitative Component:

The questionnaire domains were grounded in Empowerment Theory [12] and Social Support Theory [13], which informed the constructs of perceptions, attitudes, and preventive practices. A validated questionnaire aligned with the SESS framework was administered pre- and post-intervention. It assessed three domains: perceptions (10 items), attitudes (10 items), and preventive practices (10 items). Items were scored on a 5-point Likert scale or as dichotomous (correct/incorrect). Interpretation thresholds were ≤60% (low), 61–79% (moderate), and ≥80% (high). Content validity was confirmed by an expert panel (IOC = 0.82). Internal consistency reliability from a pilot test with 30 non-participating students yielded Cronbach’s α coefficients of 0.82 (perceptions), 0.80 (attitudes), and 0.84 (practices). Identical instruments were used pre- and post-intervention. To minimize social desirability bias, questionnaires were administered anonymously without teachers present.

Qualitative Component:

Semi-structured focus groups with peer leaders and stakeholders explored perspectives on sexual health, peer roles, and model implementation. Sessions were audio-recorded, transcribed, and thematically analyzed by two independent researchers, with triangulation and member checking to enhance credibility.

#### 2.4.3. Outcome Ascertainment for Pregnancies and STIs

Pregnancy and STIs outcomes were ascertained by standardized, anonymous self-report at follow-up during January–May 2022. Students were asked whether they had (i) a confirmed pregnancy or (ii) a clinician-diagnosed STIs within the past six months. No biological testing or record linkage was performed; therefore, these outcomes are considered exploratory and may be affected by under-reporting. Given the low base rate of such events, statistical power was limited, and observed counts are reported descriptively.

#### 2.4.4. Process Evaluation

Implementation of the SESS model was monitored using multiple indicators of dose, fidelity, and participant engagement. Attendance records were maintained for all classroom and workshop sessions to document the dose delivered versus the dose planned. Facilitator checklists and periodic observations were used to assess adherence to the intervention protocol.

Digital engagement was tracked through platform analytics, including the number of active users per week, frequency of posts, comments, and average time spent on Facebook and LINE platforms. These data were used to assess program reach and sustained participation, consistent with prior studies demonstrating the role of social media and digital platforms in adolescent health promotion [11,16,17].

Qualitative feedback from peer leaders and facilitators was also collected to capture adaptations made during delivery and perceived barriers to fidelity. A Appendix A summarizes key performance indicators (KPIs) for program dose, fidelity, and digital engagement.

### 2.5. Data Analysis

Quantitative data were analyzed using SPSS (version 26.0; IBM Corp., Armonk, NY, USA). Descriptive statistics summarized participant characteristics. Post-intervention differences were estimated using ANCOVA (SPSS version 26.0; IBM Corp., Armonk, NY, USA), adjusting for baseline values. Analyses were exploratory and allocated at the school level (two clusters). Effect sizes with 95% confidence intervals (CIs) were reported, and *p*-values were treated as descriptive. Standardized mean differences (SMDs) were also calculated to aid interpretation. Given the unit of allocation (two schools), inferential analyses were treated as exploratory, consistent with recommendations for nonrandomized evaluations [18].

Qualitative data were analyzed thematically by two trained researchers in adolescent health. Both independently coded transcripts and developed a shared coding framework through consensus discussions. To enhance trustworthiness, investigator triangulation and member checking were applied, and discrepancies were resolved through peer debriefing, in line with criteria outlined by Nowell et al. [19].

### 2.6. Ethical Considerations

Ethical approval was obtained from the Mahasarakham University Human Research Ethics Committee (Ref: 366-150/2564). Written informed consent was obtained from parents/guardians, and assent was obtained from adolescents after providing complete study information. To safeguard privacy, questionnaires were completed anonymously, without teachers present, and responses were identified only with coded numbers. Ethical safeguards followed international standards for research involving minors and global standards for adolescent-friendly health services [15]. Participation was voluntary, and no monetary incentives were offered; students only received small school supplies consistent with routine educational activities. Adolescents who experienced distress were referred to school counselors or youth-friendly health services.

## 3. Results

The results are presented in three phases, consistent with the research and development framework. Phase 1 identified contextual factors influencing adolescent sexual health. Phase 2 described the development and pilot testing of the SESS model. Phase 3 evaluated outcomes in a two-school quasi-experimental pre–post design with a non-equivalent comparison group.

### Participant Characteristics

A total of 240 adolescents aged 15–19 years participated (120 intervention, 120 comparison). Table 1 presents baseline characteristics. No meaningful differences were observed between groups in age, gender, household income, living situation, prior sex education, or parental cohabitation, indicating baseline comparability. There were no significant differences between the two groups in terms of age, gender, household income, living situation, prior exposure to sex education, or parental cohabitation (*p* > 0.05), indicating the comparability of both groups before the intervention.


*Phase 1: Situational Analysis*


The development of the SESS model was informed by a formative case–control study conducted by our research team among adolescent girls aged 15–19 years in Northeastern Thailand. The study identified several risk factors contributing to unintended pregnancy, including low levels of sexual health literacy, poor parent–child communication, peer pressure, and reliance on inaccurate information from social media. Conversely, protective factors included strong family support and awareness of sexual and reproductive rights. These findings were instrumental in shaping the core components of the model, particularly in emphasizing empowerment, social support, and digital engagement strategies tailored to the local sociocultural context.


*Phase 2: Model Development and Implementation*


This phase applied an action research approach spanning 16 weeks, focused on designing and piloting the SESS model in collaboration with school and community stakeholders. Two main components were developed and implemented:


*Development of Social Networks and Support Systems*


School-based support networks were formed, comprising educators, public health officers, school nurses, and student representatives. These networks co-developed strategies to promote adolescent sexual health within the school setting.

An online platform was co-managed by teachers, students, and researchers to serve as a hub for accurate information, interactive learning, and peer-to-peer support. This platform complemented in-person activities and allowed for extended engagement beyond the classroom.


*Empowerment-Oriented Curriculum and Activities*


Based on a situational analysis and stakeholder input, a curriculum was co-designed, focusing on four key domains: comprehensive sexuality education, pregnancy prevention, STI awareness, and rights-based knowledge.

Activities included role-playing, group reflection, scenario-based exercises, and problem-solving workshops to strengthen adolescents’ self-efficacy and decision-making skills.

Peer leaders were trained and assigned facilitation roles to enhance ownership and build sustainability. These processes emphasized student-centered learning and inclusivity.

Evaluation tools were embedded throughout this phase, enabling real-time adjustments and preparing for subsequent assessments. Data from both participants and facilitators were collected to inform Phase 3.


*Phase 3: Evaluation and Follow-up*


Following the 16-week implementation of the SESS model, outcomes were evaluated using a two-school quasi-experimental pre–post design with a non-equivalent comparison group. Both quantitative and qualitative data were collected.


*Quantitative Evaluation*


Exploratory analyses compared pre- and post-intervention scores between the intervention and comparison schools, with baseline adjustment using ANCOVA. Effect sizes and 95% confidence intervals (CIs) are reported, and *p*-values are presented descriptively given the allocation of one cluster per arm.The intervention group’s perception scores increased from 18.33 ± 7.69 at baseline to 32.00 ± 6.19 post-intervention, while the comparison group remained stable (17.80 ± 9.67 to 18.26 ± 9.84). The adjusted mean difference was 13.74 (95% CI: 10.51 to 16.97; descriptive *p* < 0.001), indicating higher post-intervention perceptions in the intervention group.Attitude scores showed only minimal change in the intervention group (111.89 ± 12.32 to 113.53 ± 11.83) compared with the comparison group (94.25 ± 5.52 to 94.58 ± 5.67). The adjusted mean difference was 18.95 (95% CI: 16.85 to 21.05; descriptive *p* < 0.001). Although statistically notable, the small absolute change may suggest a ceiling effect or response shift bias.

Practice scores in the intervention group were 71.03 ± 7.04 at baseline and 64.58 ± 7.77 post-intervention, compared with 63.25 ± 8.23 and 63.01 ± 8.16 in the comparison group. The adjusted mean difference was 1.57 (95% CI: −0.56 to 3.70; descriptive *p* = 0.148), suggesting no meaningful difference between groups at follow-up. (See Table 2).


**
*Qualitative Evaluation*
**


Focus group discussions with 20 stakeholders (students, teachers, health professionals, and administrators) were conducted to evaluate the model’s acceptability, feasibility, and perceived impact.


*Thematic analysis revealed four key findings:*
Inadequate Sexual Knowledge: Existing curricula failed to address practical aspects of reproductive health.Limited Communication: Adolescents faced barriers in discussing sexual issues with adults, relying instead on peers and social media.Need for Confidential Support: The absence of private counseling areas and trained personnel hindered students from seeking help.Positive Peer-led Engagement: Peer learning and experiential activities were perceived as effective in promoting behavior change.


These qualitative insights provided context for the quantitative improvements and informed recommendations for scaling and sustaining the SESS model in school settings.

Table 3 presents a summary of outcomes achieved across three strategic levels of the SESS Model implementation. At Level 1, a collaborative network and clear operational framework were successfully established. Level 2 emphasized the creation of a supportive school environment through policy development, increased accessibility of services, and peer-led networks. Level 3 highlighted behavioral outcomes, including a reduction in teenage pregnancies and an improvement in student awareness and preventive behaviors.

## 4. Discussion

This study evaluated the SESS model using a two-school quasi-experimental design. The results showed improvements in adolescents’ perceptions of sexual health, minimal change in attitudes, and no meaningful difference in preventive practices. These findings suggest that while knowledge-related domains are responsive to multi-component interventions, more complex constructs such as attitudes and behaviors may require more prolonged exposure and broader structural support.

### 4.1. Interpretation of Outcomes

The increase in perception scores is consistent with systematic reviews reporting that multi-component, theory-driven programs enhance adolescent health literacy and sexual health knowledge [10,11]. Attitudinal changes were limited, likely due to high baseline scores and cultural norms, a phenomenon also noted in studies of school-based sexuality education in Thailand and elsewhere [12,13]. Preventive practices did not differ between groups, echoing prior evidence that short-term interventions rarely shift behaviors unless supported by service access and environmental change [14].

### 4.2. Process Evaluation

Implementation fidelity was high, with nearly all sessions delivered as planned and the facilitator demonstrating strong adherence. Attendance was satisfactory but declined during exam weeks, and engagement on digital platforms tapered after the initial phase. Similar patterns have been documented in technology-assisted interventions, where novelty effects diminish over time [15,16,17,20]. These findings support the feasibility of implementing SESS but underscore the importance of strategies to maintain digital engagement and ensure program sustainability.

### 4.3. Limitations

This evaluation has important limitations. With only two clusters, results must be interpreted as exploratory, not confirmatory [21]. Reliance on self-reported outcomes raises the possibility of social desirability bias, particularly for sensitive topics, and rare outcomes such as pregnancy and STIs were underpowered and reported descriptively. Declining online participation also suggests that sustained engagement strategies are needed. Finally, generalizability is limited to urban and semi-urban schools in Northeastern Thailand.

### 4.4. Implications and Future Directions

Despite these limitations, the SESS model illustrates the potential of integrating system coordination, empowerment, and digital engagement within school-based sexuality education. The observed improvements in perceptions highlight the value of combining classroom and online learning. However, the limited changes in attitudes and behaviors reinforce the need for longer-term, multi-site trials with greater statistical power, objective outcome measures, and integration with youth-friendly health services. Incorporating reporting frameworks such as TREND and TIDieR [20,21] may also strengthen transparency and reproducibility in future evaluations. In addition, alignment with WHO recommendations on digital interventions and UNESCO’s international technical guidance on sexuality education can inform scalable, rights-based implementation [22,23].

## 5. Conclusions

The SESS model—a school-based, multi-component intervention grounded in systems theory, empowerment, social support, and digital engagement—was associated with improvements in adolescents’ perceptions of sexual health. Although no substantial changes were observed in attitudes or practices, the findings suggest the feasibility of combining classroom activities, peer-led initiatives, and digital engagement within existing school curricula.

The model aligns with national strategies for adolescent reproductive health and provides a practical framework for adaptation in resource-limited settings. Future research should focus on multi-site evaluations, longer-term outcomes, and integration with youth-friendly services to strengthen scalability and sustainability [15,18,21,22,23].

## Figures and Tables

**Table 1 ijerph-22-01536-t001:** Baseline Characteristics of Participants (*n* = 240).

Variable	Intervention Group (*n* = 120)	Comparison Group (*n* = 120)	*p*-Value *
Age (mean ± SD)	16.9 ± 1.2	17.0 ± 1.1	0.514
Gender: Female	75 (62.5%)	74 (61.7%)	0.873
Live with parents	95 (79.2%)	93 (77.5%)	0.692
Received sex education	55 (45.8%)	53 (44.2%)	0.843
Heard sexual health info from social media	80 (66.7%)	78 (65.0%)	0.729
Parental cohabitation	90 (75.0%)	88 (73.3%)	0.815

Note: * *p* < 0.05 was considered statistically significant.

**Table 2 ijerph-22-01536-t002:** Comparison of Mean Scores Between Intervention and Control Groups on Perception, Attitude, and Practice (*n* = 240).

Outcome Domain	Intervention Group	Comparison Group	Adjusted Mean Difference (95% CI) *	Descriptive*p*-Value
Pre-Test(mean ± SD)	Post-Test(mean ± SD)	Pre-Test(mean ± SD)	Post-Test(mean ± SD)
**Perceptions**	18.33 ± 7.69	32.00 ± 6.19	17.80 ± 9.67	18.26 ± 9.84	+13.74 (10.51 to 16.97)	<0.001
**Attitudes**	111.89± 12.32	113.53 ± 11.83	94.25 ± 5.52	94.58 ± 5.67	+18.95 (16.85 to 21.05)	<0.001
**Preventive practices**	71.03 ± 7.04	64.58 ± 7.77	63.25 ± 8.23	63.01 ± 8.16	+1.57 (−0.56 to 3.70)	0.148

Note: * Values are mean ± standard deviation (SD). Adjusted mean differences were estimated using ANCOVA, controlling for baseline scores. Analyses are exploratory given the allocation at the school level (*n* = 2 clusters). CI = confidence interval; *p*-values are descriptive.

**Table 3 ijerph-22-01536-t003:** Outcomes Across Three Strategic Levels of the SESS Model Implementation.

Level	Objective	Key Outcomes
1. Collaboration & System Readiness	Establish a collaborative task force	Sexual Health Working Group formed Clear policies & action plans Regular monitoring and referrals
2. Supportive School Environment	Promote safe spaces & access	Teen-friendly center (e.g., Condom Coffee) Peer-led networks are active Appointed youth media leaders
3. Behavioral Impact	Reduce risk behaviors	0 pregnancies post-intervention 80% showed improved knowledge & behavior 97.4% refused risky sex; 100% used protection

## Data Availability

The de-identified datasets generated and analyzed during the current study are not publicly available due to ethical restrictions involving minors. However, data may be made available from the corresponding author on reasonable request and with approval from the Mahasarakham University Human Research Ethics Committee.

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
