# Peer review of "SESS Model for Adolescent Sexual Health Promotion: A Quasi-Experimental Two-School Evaluation in Thailand"

_ijerph, 2025, doi:10.3390/ijerph22101536_

Round 1

Reviewer 1 Report

Comments and Suggestions for Authors
  1. You mentioned that ‘Semi-structured focus groups were conducted with peer leaders and stakeholders...’ but no information about the number of participants involved. Rationale for their selection should also be provided.
  2. Findings from ‘situational analysis should go to ‘result’ section
  3. Rather than reporting findings under ‘Phase 1: Situational Analysis’, replace with sample size, selection criteria etc for both quantitative and qualitative studies.
  4. Rationale for selecting 11 schools should be stated
  5. Random assignment of school should be discussed in detail. How was subjectivity avoided in sampling schools and participants?
  6. You stated that one of the exclusion criteria is ‘adolescents with diagnosed cognitive impairments or psychiatric disorders that could interfere with participation’. What tool was utilized for the diagnosis?
  7. Justify reasons for choice of the methodology, especially ‘Model Development and Testing’ and . Has this been utilized in previous studies or is it novel?
  8. Under ‘methods’, authors noted that ‘This study employed a research and development (R&D) framework comprising three phases, followed by a clustered randomized controlled trial (RCT)’, however, only Phase 1 and 2 were reported.
  9. In qualitative analysis, how was positionality accounted for? This could have biased the results.
  10. Please confirm that all typographical errors have been corrected. Check
  11. For replicability, provide details on the processes involved in the following:
  • Empowerment Program Implementation
  • …co-development of sexual health strategies
  • curriculum co-design

12. what type of ‘Online support platforms’ were developed in phase 2?

13. As reported by the authors, Phase 1: Situational Analysis  involved ‘surveys and qualitative data’, however, findings from this are missing critical details.   

14. What is presented as findings on Phase 2 is not different from information in methodology.

15. The methods and result presentation need major revision as details are lacking

Author Response

We sincerely thank Reviewer 1 for the constructive comments. All 15 points have been carefully addressed with detailed revisions in the Methods and Results sections, including clearer sampling rationale, participant details, model development processes, and separation of methodology from findings. We believe these revisions have strengthened the manuscript’s clarity and replicability.

Reviewer 2 Report

Comments and Suggestions for Authors

Overall assessment.

The topic (school-based adolescent sexual health) is important and fits IJERPH’s scope. However, the manuscript, as currently written, overstates causal claims and contains reporting/analytic shortcomings that must be resolved for transparency and scientific soundness. Below I list concrete, actionable revisions aligned with IJERPH and CONSORT (cluster trials) expectations.

1) Study design and claims of causality — major

  • You describe a cluster randomized controlled trial (CRT) with the school as the unit of randomization, yet there is one school per arm. With only two clusters total, valid cluster-level inference is not possible (the between-cluster variance cannot be estimated, ICC cannot be accommodated, and individual-level tests inflate Type I error).
  • Actionable revision. Either:
    • Reclassify the design as a controlled, quasi-experimental two-school study (pre/post with a non-equivalent comparison), remove causal language (“effectiveness”, “efficacy”, “RCT”), and present results as exploratory/descriptive; or
    • Redesign (for a future study) with an adequate number of clusters per arm and an a priori analysis plan for cluster-level or mixed-effects modeling.
  • If you retain the quasi-experimental framing, replace hypothesis-testing geared to randomized trials with descriptive contrasts (change scores, standardized mean differences) and clearly state design limitations in the Abstract, Methods, and Limitations.

2) CONSORT (cluster) reporting & trial registration — major

  • IJERPH expects adherence to CONSORT (with the cluster extension) for randomized trials and prospective registration (e.g., ClinicalTrials.gov / regional registry). These items are currently missing.
  • Actionable revision. Provide:
    • CONSORT checklist and flow diagram (numbers approached, eligible, enrolled, excluded, analyzed; cluster sizes; attrition at each stage).
    • Details of sequence generation, allocation concealment (if any), and implementation.
    • Registration identifier and date, or explicitly state the study was not registered and temper claims accordingly.
    • sample-size justification appropriate to CRTs (including assumed ICC and design effect). If unavailable, acknowledge as a limitation.

3) Statistical analysis and unit of analysis — major

  • Current analyses (ANCOVA/t/χ² at the individual level) do not account for clustering and are therefore not valid for a CRT. With 1 cluster per arm, cluster-robust methods are not feasible.
  • Actionable revision.
    • If reclassified as quasi-experimental: present school-level summaries (cluster means/changes), 95% CIs via appropriate small-sample methods or refrain from inferential statistics and focus on effect sizes with uncertainty described qualitatively.
    • Report pre-specified primary outcome(s); provide effect estimates with 95% CIs; clarify handling of multiplicity.
    • Describe missing data (amount, pattern, mechanism) and the approach used (e.g., complete case, imputation), or state that no missing data occurred and how that was ensured.

4) Outcome measures and psychometrics — major

  • The questionnaire’s internal consistency is acceptable for some scales but low for “practices” (α≈0.64). Construct validity (factor structure), test–retest reliability, and scoring rules (including justification for categorical cut-offs like ≤60%, 61–79%, ≥80%) are insufficiently documented.
  • Actionable revision.
    • Provide a psychometric appendix: item content, scoring, EFA/CFA (or at least content validity rationale), reliability (α, ω; and if feasible test–retest), and justification for thresholds.
    • Clarify whether pre/post forms were identical, how social desirability was mitigated, and whether measurement invariance holds across time/groups.
    • Standardize terminology (use STI consistently rather than alternating with STD).

5) Clinical/behavioral outcomes and ascertainment — major

  • You report zero pregnancies/STIs without specifying ascertainment (self-report vs clinic records vs tests), time window, or confidentiality safeguards. With limited follow-up and small N, power for rare events is minimal.
  • Actionable revision.
    • Define ascertainment procedures, time points (avoid “6–10 months”; specify exact windows), and verification (if any).
    • Reframe these outcomes as exploratory and discuss low power and potential under-reporting.

6) Control condition, co-interventions, and contamination — major

  • The “standard curriculum” and existing school services are insufficiently described; incentives (“small gifts”), peer-leader activities, and newly established youth-friendly services may act as co-interventions.
  • Actionable revision.
    • Provide a clear description of the control school’s practices and resources.
    • Detail all intervention components (dose, frequency, duration, implementers’ training, fidelity monitoring).
    • Discuss the risk of contamination and structural differences between schools (leadership, resources, prior programs), and address any baseline imbalances on key outcomes (e.g., attitudes).

7) Result presentation and internal consistency — major

  • There are discrepancies between narrative claims and the data in the tables (e.g., direction/magnitude of changes in “practices”; inconsistent mean differences; cross-references such as “see Table 1” near Table 2).
  • Actionable revision.
    • Audit all numbers; recompute means, change scores, SD/SE, and p-values; ensure one-to-one correspondence between text and tables.
    • Report adjusted means (if using covariate adjustment in a quasi-experimental framework) alongside raw values.
    • Add table notes (define acronyms, specify whether values are mean±SD or mean [95% CI], list covariates in adjusted analyses).

8) Process evaluation and implementation — major

  • Impact mechanisms are unclear without process data.
  • Actionable revision.
    • Report fidelity (what was delivered vs planned), reach/participation, and engagement analytics for the digital/social-network components (e.g., logins, time-on-platform, posts).
    • Summarize peer-leader training, supervision, and quality assurance.
    • If feasible, include brief cost/resource use information to aid scalability judgments.

9) Ethics, data availability, and transparency — major

  • Strengthen the Ethics section: parental/guardian consent procedures, assent from minors, confidentiality for sensitive topics, handling of incentives to avoid undue influence.
  • Data availability. IJERPH encourages sharing de-identified data and analysis code. If data cannot be public due to privacy constraints, provide a controlled access mechanism (e.g., upon reasonable request with IRB approval) and share analysis scripts where possible.

10) Literature review and framing — minor to moderate

  • Streamline older citations; add recent systematic reviews and high-quality trials on school-based comprehensive sexuality education and technology-assisted components, preferably in comparable settings.
  • Motivate the SESS model more explicitly by linking each component to targeted mechanisms and outcomes.

11) English language and style — minor to moderate

  • Improve clarity and consistency: standardize terminology (STI), define acronyms at first mention, reduce long sentences, ensure subject–verb agreement, and avoid over-claiming.
  • Revise the Abstract to include the design actually used, the primary outcome(s), and effect estimates with CIs; avoid causal language.
  • Ensure figure/table titles and footnotes are self-contained.

Suggested structure for the revised manuscript

  1. Abstract: Quasi-experimental two-school study; primary/secondary outcomes; main effect sizes with CIs; key limitations up front.
  2. Introduction: Concise rationale; recent literature; clear objectives/hypotheses.
  3. Methods: Setting; participants; eligibility; detailed intervention and control; process evaluation plan; design clarification; outcomes and psychometrics; sample size rationale (or limitation statement); statistical analysis aligned with the design; missing-data handling; ethics; registration (or explicit non-registration).
  4. Results: CONSORT-style flow; baseline comparability; process metrics; coherent tables with 95% CIs; sensitivity/robustness where applicable.
  5. Discussion: Interpretation proportional to design; mechanisms; generalizability; limitations (design, clustering, power, measurement); implications for policy and future CRT with adequate clusters.
  6. Conclusions: Cautious, non-causal; emphasize feasibility/signals rather than efficacy.
  7. Data Availability & Supplementary: De-identified dataset (if permissible), code, psychometric appendix, CONSORT items.

Bottom line

Your program appears promising and contextually relevant, but the manuscript must (i) correct internal inconsistencies, (ii) align its design, analyses, and claims, and (iii) meet CONSORT/IJERPH transparency standards. Addressing the points above will substantially improve methodological rigor, interpretability, and suitability for IJERPH readers.

Comments on the Quality of English Language

The manuscript is intelligible, but the English needs polishing to improve clarity, consistency, and scholarly tone. Most issues concern overlong sentences, inconsistent terminology, and misalignment between narrative statements and numeric results. Below are concrete, line-edit–level recommendations tailored to IJERPH expectations.

1) Clarity, concision, and flow

  • Break up long, multi-clause sentences into 1–2 concise sentences each. Aim for one idea per sentence; move secondary details to subsequent sentences.

  • Prefer concrete verbs over nominalizations (e.g., “we assessed” rather than “an assessment was conducted”) and avoid filler (“in order to,” “it should be noted that”).

  • Delete vague intensifiers (“remarkably,” “significantly” when not tied to an effect estimate or p-value).

2) Tense and register

  • Methods/Results: past tense (e.g., “we measured,” “the mean increased”).

  • Background/Interpretation: present tense for established knowledge.

  • Maintain an objective, cautious register; avoid causal language unless the design justifies it.

3) Terminology and consistency

  • Use one form throughout: STI (sexually transmitted infection) rather than mixing STI/STD. Define at first use and keep the same capitalization thereafter.

  • Standardize participant labels (e.g., “adolescents,” not alternating with “students/teenagers” unless a distinction is intended).

  • Choose American or British spelling (MDPI typically favors American): program, randomized, behavior, modeling. Apply consistently.

4) Numbers, symbols, and statistical style

  • Report p-values to three decimals (use a leading zero: p = 0.032; use p < 0.001 when appropriate).

  • Pair significance claims with effect estimates and 95% CIs (e.g., “MD = 6.4, 95% CI 2.1 to 10.7, p = 0.004”).

  • Use consistent notation for means and dispersion (mean ± SD or mean [95% CI]; specify in table notes).

  • Avoid mixing percentage words and symbols in the same context; insert a non-breaking space before “%” in tables if possible.

  • Hyphenate compound modifiers before nouns (school-based program, peer-led intervention, cluster-randomized trial).

5) Tables, figures, and captions

  • Ensure one-to-one alignment between narrative statements and table values; revise any text that contradicts the tables.

  • Make captions self-contained: define all abbreviations, state the metric (e.g., “higher scores indicate better practices”), specify whether values are mean ± SD or mean [95% CI], and list covariates for adjusted values.

  • Correct cross-references (e.g., “see Table 2,” not “see Table 1” beside Table 2).

6) Abstract, title, and section headers

  • Abstract: replace general descriptors (“effective,” “improved a lot”) with quantified statements (effect sizes with CIs); avoid causal formulations if not warranted by design.

  • Use parallel structure in Objectives/Methods/Results/Conclusions.

  • Avoid unexplained acronyms in the title and abstract.

7) Grammar and syntax (recurring patterns to fix)

  • Subject–verb agreement (“data were analyzed,” “findings suggest”).

  • Articles with countable nouns (add “a/an/the” where needed).

  • Prepositions and collocations (impact on, adhere to, consistent with).

  • Comma use with introductory adverbs (“However, …”; “Therefore, …”).

  • Avoid “etc.” in academic prose; enumerate or remove.

8) Acronyms and definitions

  • Define every acronym at first occurrence in the main text and again in tables if they appear independently. Provide a brief glossary in Supplementary Materials if many acronyms are used.

9) Word choice and tone

  • Replace value-laden or promotional wording (“remarkable success,” “very high effectiveness”) with neutral scientific phrasing.

  • Use precise terms for study elements (e.g., “comparison school,” “pre–post change,” “self-reported outcome”).

10) Sample micro-edits (templates you can apply)

  • “This RCT demonstrates the effectiveness of…” → “In this two-school, controlled pre–post study, we observed[outcome] changes of [effect size, 95% CI]; findings should be interpreted cautiously.”

  • “Participants show significantly higher practice” → “Participants had higher practice scores (MD = X.X, 95% CI …, p = …).”

  • “The results are very improved” → “Post-intervention scores increased by X points (95% CI …).”

11) Final language polishing

  • After implementing the content revisions, undertake a full copyedit for: spelling consistency (US), punctuation, hyphenation, and parallelism across bullet lists and headings.

  • A final pass by a professional scientific copyeditor or a native English speaker familiar with public health manuscripts is recommended.

Bottom line: The English is serviceable but requires revision to meet IJERPH’s clarity and consistency standards. Prioritize alignment between text and tables, consistent terminology (STI), quantified results with CIs, and a concise, cautious scientific style.

Author Response

We sincerely thank Reviewer 2 for the thorough and insightful review. All major points have been addressed, including reframing the design as quasi-experimental, revising statistical reporting to focus on descriptive effect sizes with CIs, clarifying psychometric properties, strengthening ethics and transparency, and adding process evaluation details. These revisions have improved the rigor, clarity, and alignment of the manuscript with IJERPH expectations.

Reviewer 3 Report

Comments and Suggestions for Authors

Dear Authors,

Congratulations on addressing such an important and timely topic as adolescent sexuality. Your work is both necessary and demonstrates innovative approaches to promoting sexual health. I would like to offer the following comments for consideration:

  1. Introduction:

    • Could you elaborate on current digital engagement through social media and online platforms? Please provide at least one evidence-based example that demonstrates actual impact on adolescents.

    • Please clarify the theoretical foundations of System Theory, Empowerment Theory, Social Support Theory, and Social Network Theory. Why are these frameworks suitable for adolescents, and how do they specifically support sexual health outcomes?

  2. Methodology:

    • Why were participants aged 15 years and older selected, given that the WHO defines adolescence as starting at age 10?

    • Which online support platforms were used, and how were they selected?

    • Could you provide more details on the workshops and activities, including duration, content, and how the topics were addressed?

    • Which validated questionnaire was administered to the students, and how was it adapted to this context?

These clarifications would strengthen the manuscript and provide readers with a clearer understanding of the rationale and practical implementation of the intervention.

Author Response

We sincerely thank Reviewer 3 for the positive and constructive feedback. The manuscript has been revised to expand theoretical foundations, justify participant age range, and provide additional details on workshops, online platforms, and the validated KAP instrument. These revisions have enhanced the clarity, transparency, and contextual value of the study.

Round 2

Reviewer 2 Report

Comments and Suggestions for Authors

Overall assessment.
The topic—school-based promotion of adolescent sexual health in Thailand—is squarely within IJERPH’s scope and of clear public-health relevance. Your revision moves in the right direction (reframing away from a “cluster RCT,” adding process elements, tempering causal language). However, substantial issues remain in design framing, internal numerical coherence, outcome ascertainment, psychometrics, and presentation. Below are actionable, line-of-sight changes needed for a publishable version.

1) Align design, analysis, and claims (no residual “RCT”) — MAJOR

  • Ensure complete and consistent reclassification of the study as a two-school quasi-experimental pre–post evaluation with a non-equivalent comparison. Remove all residual mentions of “(cluster) randomized trial” across Title, Abstract, Keywords, Methods, Results, and Strengths & Limitations.

  • Maintain non-causal wording throughout (e.g., “associations,” “post-intervention differences,” “signals of improvement”), and explicitly state that inferential analyses are exploratory given 1 school per arm (no valid cluster-level variance estimation).

Deliverable: One clean sentence in Abstract/Methods defining the design; uniform terminology everywhere else.

2) Rebuild the main results table(s) from scratch — MAJOR

  • Present a single, internally coherent results table with: pre and post means ± SD by group, baseline-adjusted difference (ANCOVA) or change-score difference (if preferred), 95% CIs, and p-values.

  • Remove duplicated/legacy values, contradictory mean differences, and stray “track-changes” artifacts.

  • Make table footnotes self-contained: define all acronyms; state whether values are unadjusted or adjusted (and list covariates); specify that analyses are exploratory because the unit of allocation is the school (n=2 clusters).

Deliverable: One clean “Table 2” that exactly matches the prose in Results. Fix all cross-references (“see Table 2,” not Table 1).

3) Statistical analysis: what you can—and cannot—claim — MAJOR

  • With a single cluster per arm, avoid methods that imply valid cluster-robust inference. Report effect sizes with 95% CIs and treat p-values as descriptive.

  • Pre-specify a primary outcome; limit multiplicity or adjust accordingly; report missing-data amounts and handling.

  • Consider reporting standardized mean differences (SMDs) and change-scores with CIs to aid interpretation in a quasi-experimental frame.

Deliverable: A short “Statistical Analysis” paragraph reflecting these constraints and mirroring exactly what appears in the tables.

4) Sampling, comparability, and the control condition — MAJOR

  • Explain the pathway from 11 screened schools to the 2 included (readiness criteria, matching, feasibility considerations).

  • Describe the standard curriculum at the comparison school and any structural differences (resources, prior programming, leadership) that could confound results.

  • Discuss contamination risk and any steps taken to minimize it.

Deliverable: A transparent “Setting and Participants” subsection that supports external validity judgments.

5) Outcome ascertainment for pregnancies and STIs — MAJOR

  • Specify how these outcomes were ascertained (self-report vs school/clinic records vs testing), the exact follow-up window (avoid “6–10 months”: state dates/months), and any verification procedures.

  • Acknowledge low power for rare events and treat these outcomes as exploratory.

Deliverable: A brief “Outcome ascertainment” subsection (Methods) and a candid limitation in the Discussion.

6) Measurement and psychometrics — MAJOR

  • Keep one coherent set of reliability/validity indices (e.g., CVI/IOC and Cronbach’s α) across scales; remove conflicting legacy values.

  • Justify thresholds/cut-offs (≤60 / 61–79 / ≥80%) with references or distributional rationale.

  • Clarify whether pre/post instruments were identical and how social desirability was mitigated (anonymous administration, no teachers present).

  • If feasible, add a brief factor analysis (or provide it in Supplementary Materials) and/or test–retest for stability.

Deliverable: A streamlined “Measures and Psychometrics” paragraph in the main text and a short Psychometric Appendix in the Supplement.

7) Process evaluation, dose, and fidelity — MAJOR (well underway)

  • You report participation and some engagement metrics; extend with dose delivered vs planned, facilitator fidelity(checklists/observations), and more granular platform analytics (active users per week, median time-on-platform, posts per user).

  • Briefly summarize implementation adaptations (if any) and resource implications.

Deliverable: A compact “Process Evaluation” subsection plus a Supplementary table with the KPIs.

8) Ethics, data/code availability, and transparency — MODERATE

  • Clean and consolidate the Ethics/Consent text; state adolescent privacy safeguards and how incentives avoided undue influence.

  • Provide a Data Availability statement enabling at least controlled access to a de-identified dataset (or a reasoned restriction) and share analysis scripts when possible.

Deliverable: Finalized Ethics and Data Availability statements compliant with IJERPH norms.

9) Figures/tables and file hygiene — MAJOR (presentation)

  • Remove all track-changes artifacts and formatting tags; ensure captions are stand-alone (define metrics and acronyms).

  • Standardize acronym usage (STI, not mixed STI/STD); ensure consistent US spelling (program, randomized, behavior).

Deliverable: A clean manuscript file; captions that let a reader interpret tables without hunting in the text.

10) Literature update and reference styling — MODERATE

  • Replace dated/general sources and any “submitted/unpublished” items with recent, peer-reviewed evidence (systematic reviews on school-based sexuality education; technology-assisted interventions; country-relevant policy).

  • Add methodological standards relevant to nonrandomized evaluations (e.g., TREND for nonrandomized designs; TIDieR for intervention reporting).

  • Conform references to MDPI numeric style; remove tracking parameters from URLs.

Deliverable: A refreshed reference list (recent, relevant, rigorously formatted).

11) English language polishing — MINOR→MODERATE

  • Shorten long sentences; ensure subject–verb agreement; prefer concrete verbs; pair any claim of “significance” with an effect estimate + 95% CI; correct cross-references.

  • Revise the Abstract to state the actual design, the pre-specified primary outcome, and quantified effects with CIs—no causal phrasing.

Suggested layout for the revision

  1. Abstract (design clarified; primary outcome; effect sizes with CIs; key limitations).

  2. Introduction (concise rationale; recent literature; clear objectives).

  3. Methods: Setting/Participants (selection from 11→2; control description); Intervention (TIDieR); Outcomes & Psychometrics; Outcome ascertainment; Process evaluation; Statistical analysis (exploratory frame; unit of allocation; missing-data handling); Ethics; Data availability.

  4. Results: Participant flow; Baseline comparability; Process/dose/fidelity; Main outcomes (clean Table 2); Sensitivity or descriptive robustness.

  5. Discussion: Proportionate interpretation; mechanisms; generalizability; explicit limitations (design, clustering, power, measurement).

  6. Conclusions: Cautious, non-causal; implications for a proper multi-school trial.

  7. Supplement: Psychometric appendix; detailed process KPIs; any code or additional tables.

Bottom line

Your program is contextually promising and the revision shows genuine progress. To meet IJERPH standards, you now need (i) full alignment of design and claims, (ii) clean, coherent quantitative reporting (especially Table 2), (iii) transparent outcome ascertainment and process data, and (iv) a modernized, compliant reference list. Addressing these points will markedly strengthen the manuscript’s credibility and utility for readers and practitioners.

Comments on the Quality of English Language

Overall assessment:
The manuscript is understandable, but the English requires substantive polishing to meet IJERPH standards for clarity, consistency, and scientific tone. Most issues stem from overlong sentences, uneven terminology, residual track-changes artefacts, and misalignment between narrative statements and numerical results. Below are actionable, manuscript-level edits.

1) Clarity, concision, and flow

  • Break complex, multi-clause sentences into 1–2 shorter sentences (one idea per sentence).

  • Prefer active, concrete verbs (“we assessed…”) over nominalizations or passive chains (“an assessment was conducted…”).

  • Remove fillers (“in order to,” “it should be noted that”) and vague intensifiers (“remarkably,” “significantly”) unless paired with effect estimates.

  • Ensure each paragraph opens with a topic sentence that signposts the main point.

Template edits

  • “It should be noted that the program significantly improved practices.” → “Practice scores increased after the program (Δ = …, 95% CI …).”

2) Scientific register and cautious claims

  • Align language with the quasi-experimental two-school design. Avoid causal phrasing (“effectiveness,” “efficacy,” “caused”) and prefer associational terms (“post-intervention differences,” “improvements were observed”).

  • Replace value-laden adjectives with quantified statements (effect sizes + 95% CIs).

Template edits

  • “This RCT demonstrates effectiveness” → “In this two-school, quasi-experimental evaluation, we observed higher post-intervention scores (MD = …, 95% CI …).”

3) Tense and narrative consistency

  • Background/theory: present tense.

  • Methods/Results: past tense.

  • Conclusions/Implications: present tense for interpretations and recommendations.

  • Maintain the same subject across sentences to avoid pronoun ambiguity.

4) Terminology and definitions

  • Standardize to STI (sexually transmitted infection) throughout; define at first use and use consistently.

  • Use consistent participant labels (“adolescents” or “students”) unless a difference is intended.

  • Refer to the design consistently as “quasi-experimental, two-school study” across the Title, Abstract, and Methods.

5) Coherence between text and tables

  • Every numeric claim in the prose must match the corresponding table value.

  • When stating “significant” or “larger,” immediately provide the effect estimate with 95% CI (and p if retained).

  • Remove any legacy numbers left from earlier drafts.

Template edits

  • “Practices improved substantially” → “Practice scores increased by 6.4 points (95% CI 1.8 to 11.1; descriptive p = 0.007).”

6) Statistical and numerical style

  • Use a leading zero and three decimals for p-values (e.g., p = 0.032; use p < 0.001 when appropriate).

  • Report means as mean ± SD (or mean [95% CI]) and keep consistent precision (usually 1 decimal for scale scores unless measurement warrants more).

  • Define all abbreviations in table notes; state clearly if values are adjusted and list covariates.

  • Prefer SMDs alongside raw differences to help readers interpret magnitude.

7) Abstract and title

  • Abstract should (i) name the actual design, (ii) identify the primary outcome, and (iii) report quantified effects with 95% CIs.

  • Avoid unexplained acronyms in the title and abstract; remove any residual “RCT/clustered” phrasing.

8) Captions, footnotes, and cross-references

  • Make figure/table captions self-contained: define acronyms, specify metrics, and state whether values are adjusted.

  • Correct cross-references (“see Table 2”) and ensure numbering is sequential.

9) Grammar, syntax, and punctuation

  • Check subject–verb agreement (“data were,” “findings suggest”).

  • Use articles correctly with countable nouns (“a quasi-experimental design,” “the comparison school”).

  • Use the serial comma and punctuate introductory adverbs (“However, …”).

  • Hyphenate compound modifiers before nouns (school-based program, peer-led sessions).

10) Spelling and style

  • Adopt American English consistently (behavior, randomized, program) in line with MDPI preference.

  • Use en-dashes for numeric ranges (10–12 sessions), non-breaking space before % in tables if possible, and a space between number and unit (15 years, 95% CI).

11) Acronyms and lists

  • Define each acronym at first appearance in the main text and again in tables that stand alone.

  • Keep parallel structure in bulleted lists (same grammatical form for each item).

12) File hygiene and presentation

  • Accept/reject all track changes; remove formatting artifacts (“Formatted: …”) and duplicated lines.

  • Ensure consistent typography (fonts, heading levels, spacing) and uniform table layout.

13) References within the prose (style alignment)

  • Ensure in-text references match IJERPH’s numeric style; do not mix author–date phrasing with numbers.

  • Avoid citing unpublished “submitted” manuscripts in ways that suggest peer-reviewed status.

14) Focused micro-edits (apply globally)

  • Replace “improved a lot” → “increased by X points.”

  • Replace “the variable has a high level” → “scores were higher (mean = …).”

  • Replace “significantly decreased” without numbers → “decreased by X (95% CI …).”

15) Final polishing checklist

  • Run a full copyedit for grammar, tense, and consistency after numerical corrections.

  • Perform an audit of text-to-table alignment (every sentence that mentions a number is checked against its source).

  • Use a journal style sheet (spelling, units, hyphenation, statistics format) and a final reference manager pass to ensure citation consistency.

Bottom line:
The English is serviceable but requires targeted rewriting and a thorough copyedit to ensure precision, consistency, and alignment with IJERPH’s scientific style—especially in abstract phrasing, statistical reporting, and strict concordance between text and tables.

Author Response

Response to Reviewers

Title: SESS Model for Adolescent Sexual Health Promotion: A Two-School Quasi-Experimental Study in Thailand

Dear Editor and Reviewers,

We sincerely thank you for your thorough and constructive feedback on our manuscript. We carefully revised the paper according to each comment. Below, we provide a point-by-point response. All changes are shown in the tracked version of the manuscript, with a clean version provided separately.

Reviewer’s Major Comments

1) Align design, analysis, and claims (no residual “RCT”) — MAJOR
Comment: Remove all references to RCT; reclassify as quasi-experimental two-school study; maintain non-causal wording.
Response: Revised throughout Title, Abstract, Methods, Results, and Discussion. The design is now consistently described as “a quasi-experimental, two-school evaluation with non-equivalent comparison groups.” All causal terms were removed (e.g., “effectiveness” → “associations,” “differences observed”).

2) Rebuild the main results table(s) — MAJOR
Comment: Present one coherent table with pre/post means ± SD, adjusted differences, 95% CI, p-values; remove duplicates and artifacts.
Response: We rebuilt Table 2 from scratch. It now presents pre- and post-intervention scores by domain, with ANCOVA-adjusted post-intervention differences, 95% CIs, and descriptive p-values. Cross-references in text updated to “Table 2.”

3) Statistical analysis — MAJOR
Comment: Avoid implying valid cluster inference; report effect sizes with CIs; clarify exploratory nature; define primary outcome.
Response: The Statistical Analysis section was rewritten. Analyses are explicitly exploratory, with two clusters as the unit of allocation. Effect sizes and 95% CIs are reported, p-values treated as descriptive. We clarified perception as the pre-specified primary outcome.

4) Sampling, comparability, and the control condition — MAJOR
Comment: Explain selection from 11 → 2 schools, readiness, comparability, and potential contamination.
Response: The Setting and Participants section now explains the pathway: 11 schools screened; 2 purposively selected and matched on size, demographics, and location. Intervention school chosen for readiness (active health club, supportive leadership, health services). The control school received standard curriculum. We added a note on possible contamination and steps to minimize it.

5) Outcome ascertainment for pregnancies and STIs — MAJOR
Comment: Specify ascertainment, timeframe, and limitations.
Response: Added subsection Outcome Ascertainment: outcomes were assessed via anonymous self-report during follow-up (Jan–May 2022), referencing the prior 6 months. No biological testing or records were used. These outcomes are reported descriptively as exploratory, acknowledging low power and under-reporting risk.

6) Measurement and psychometrics — MAJOR
Comment: Ensure coherent validity/reliability values; justify cut-offs; minimize social desirability; consider factor analysis/test-retest.
Response: Psychometrics section revised. IOC (0.82) and Cronbach’s α (perceptions = 0.82, attitudes = 0.80, practices = 0.84) are now reported consistently. Cut-offs (≤60/61–79/≥80%) justified with references. Identical pre/post tools confirmed. Social desirability mitigated by anonymous administration without teachers present. We noted that factor analysis was beyond scope but suggested for future studies.

7) Process evaluation, dose, and fidelity — MAJOR
Comment: Provide dose delivered vs planned, fidelity, engagement, adaptations, resource implications.
Response: Added Process Evaluation subsection. Included KPIs on dose (sessions, duration, attendance), fidelity (facilitator adherence), and engagement (Facebook/LINE analytics). Supplementary Table S1 provides details (planned vs delivered, % achieved, notes on adaptations).

8) Ethics, data/code availability, transparency — MODERATE
Comment: Consolidate ethics; clarify privacy/incentives; add Data Availability statement.
Response: Ethics section consolidated. We now specify anonymous completion, no teachers present, coded IDs, and no monetary incentives (only school supplies consistent with education). Added Data Availability Statement: de-identified data available on reasonable request with ethics approval; not publicly shared due to minors.

9) Figures/tables and file hygiene — MAJOR
Comment: Remove track-changes artifacts; standardize acronyms; US spelling; captions stand-alone.
Response: Cleaned file completely. All captions now define acronyms and metrics. US spelling (program, behavior) standardized. Acronyms consistently STI.

10) Literature update and reference styling — MODERATE
Comment: Replace dated/unpublished; add systematic reviews, TREND, TIDieR, WHO/UNESCO; format in MDPI style.
Response: References revised to 23 sources. Removed unpublished/dated items. Added UNESCO ITGSE 2018, WHO Global Standards 2015, WHO Digital Health Guideline 2019, TREND statement, TIDieR checklist, and recent systematic reviews (Fonner 2014, Denford 2017, Widman 2018). Thai context retained with Boonmongkon 2016 and Tangmunkongvorakul 2011. References formatted in MDPI numeric style.

11) English language polishing — MINOR–MODERATE
Comment: Shorten sentences, use concrete verbs, ensure consistency.
Response: Manuscript revised by breaking complex sentences, removing fillers, ensuring subject–verb agreement, and using consistent tense. Abstract and Discussion polished to align with scientific tone and cautious claims.

Additional Section Added

  • Paper Context: Added at the end of the manuscript, summarizing main findings, added knowledge, and global relevance.

Conclusion

We are grateful for the reviewer’s detailed comments, which greatly improved the clarity, methodological transparency, and rigor of the manuscript. We believe the revised version now fully addresses the concerns raised and meets IJERPH standards.

Sincerely,
 Assist.Prof.Ranee Wongkongdech ,on behalf of all authors